# Comparison of Different Invasive and Non-Invasive Methods to Characterize Intestinal Microbiota throughout a Production Cycle of Broiler Chickens

**DOI:** 10.3390/microorganisms7100431

**Published:** 2019-10-10

**Authors:** Jannigje G. Kers, Egil A.J. Fischer, J. Arjan Stegeman, Hauke Smidt, Francisca C. Velkers

**Affiliations:** 1Faculty of Veterinary Medicine, Department of Farm Animal Health, Utrecht University, 3584 CL Utrecht, The Netherlandsj.a.stegeman@uu.nl (J.A.S.); f.c.velkers@uu.nl (F.C.V.); 2Laboratory of Microbiology, Wageningen University & Research, 6708 WE Wageningen, The Netherlands; Hauke.Smidt@wur.nl

**Keywords:** microbiome, 16S rRNA, methods, gut, avian, poultry

## Abstract

In the short life of broiler chickens, their intestinal microbiota undergoes many changes. To study underlying biological mechanisms and factors that influence the intestinal microbiota development, longitudinal data from flocks and individual birds is needed. However, post-mortem collection of samples hampers longitudinal data collection. In this study, invasively collected cecal and ileal content, cloacal swabs collected from the same bird, and boot sock samples and cecal droppings from the litter of the broilers’ poultry house, were collected on days 0, 2, 7, 14 and 35 post-hatch. The different sample types were evaluated on their applicability and reliability to characterize the broiler intestinal microbiota. The microbiota of 247 samples was assessed by 16S ribosomal RNA gene amplicon sequencing. Analyses of α and β measures showed a similar development of microbiota composition of cecal droppings compared to cecal content. Furthermore, the composition of cecal content samples was comparable to that of the boot socks until day 14 post-hatch. This study shows that the value of non-invasive sample types varies at different ages and depends on the goal of the microbiota characterization. Specifically, cecal droppings and boot socks may be useful alternatives for cecal samples to determine intestinal microbiota composition longitudinally.

## 1. Introduction

In the six- to eight-week life span of commercial broilers the intestinal microbiota composition changes rapidly [1,2]. Several studies have revealed associations between intestinal microbiota composition and health, and the production performance of broiler chickens [3,4,5,6,7,8], but underlying mechanisms have remained unclear. To elucidate how intestinal microbiota composition interacts with parameters related with health and performance, e.g., maturation of the immune system, growth and feed conversion, longitudinal data is of great value. To this end, monitoring of intestinal microbial development from the first week of life onwards in individual birds would be especially important. This would help to improve the research of underlying mechanisms of intestinal microbiota succession and may facilitate new interventions to increase health and performance of broiler chickens [9,10,11]. The added value of longitudinal data has also recently been demonstrated in humans, where daily sampling revealed that similar foods had different effects on the intestinal microbiota in different individuals and that the variation of the microbiota depends on at least two days of dietary history [12].

The sample type has a large impact on the observed intestinal microbiota, because the diversity and functionality of the intestinal microbiota substantially differs per intestinal region in mammals and poultry [13,14,15]. In the ceca of chickens, the microbiota is quantitatively and functionally most developed compared to other parts of the chickens’ intestinal tract [16,17]. The microbiota in the cecal content has been widely investigated because its functionality, mostly fermentation, is affected by diet [15,16,18,19,20,21]. The ileum plays an important role for digestion and absorption of nutrients [16,22], and its microbial composition is more sensitive to transient shifts, and shows lower diversity and richness compared to cecal content [15,23,24,25,26]. Therefore, in broiler intestinal microbiota research cecal content samples are often of main interest, both in experimental settings and field studies. The collection of ileal and cecal content samples has to be done post-mortem, which is not only undesirable because it requires euthanasia of broilers, but also makes longitudinal sampling of individual birds impossible. Fecal droppings of chickens would facilitate longitudinal sampling, however, several studies in chickens have shown that those samples do not reflect the cecal microbiota [21,27,28]. Also, in literature about humans the use of fecal (stool) samples as a reference for the intestinal microbiota is under debate [13,29,30,31]. Those samples are easy to collect longitudinally but might be less useful to unravel functionalities of the intestinal microbiota [13,29,32]. In addition, there is individual variation in the broilers’ intestinal microbiota composition, both between as well as within studies and flocks. [33,34]. Hence, re-sampling of the same bird throughout an entire study could be advantageous in many types of microbiota research [27,35].

A non-invasive sample type used in several studies is the cloacal swab [36,37,38]. The intestinal microbiota from cloacal swabs was distinct from cecal content samples in broilers of 25 days of age but showed some overlap with ileal content samples [21]. A recent study showed that microbiota composition assessed using cloacal swabs was not reflecting the cecal content or ileal content samples but most closely represented the microbiota found in litter samples [4]. In four- to six-week-old ostriches, cloacal swabs showed similar results as fecal samples and also differed from cecal content and ileal content samples [28]. Another non-invasive sample type is cecal droppings, as in addition to fecal droppings, chickens also excrete cecal droppings [39]. The similarity of the cecal content and cecal droppings has been described before for broilers at the end of the production cycle [27], but not in broilers during the production cycle. A third non-invasive sample type are boot socks samples, which are commonly used in poultry houses to monitor the presence of *Salmonella* [40,41]. Theoretically, boot socks might also be applicable as a method to measure the microbiota of a poultry house but have not been applied in microbiota research so far. To our knowledge, there are no published reports where the potential usability of boot socks or cecal droppings was evaluated for microbiota research from shortly after hatch throughout a broiler production cycle. 

Therefore, the aim of this study is to evaluate invasively collected samples and non-invasively collected samples with regard to characterization of the intestinal microbiota composition. On two different broiler farms cloacal swabs were collected intravitally and from the same birds cecal and ileal content was collected at post-mortem at 0, 2, 7, 14 and 35 days of age. At the same time points, cecal droppings and boot socks samples were obtained from the broiler house. We compared the microbiota composition of cecal and ileal content to cloacal swabs, and compared cecal content to cecal droppings, and cecal content with boot socks samples. In addition to the microbiota composition, also the discriminative properties of the sample types were assessed, by evaluating its ability to distinguish farms or the age of the broilers. The microbiota composition of 247 samples was assessed by 16S ribosomal RNA (rRNA) gene amplicon sequencing, to determine intestinal microbiota composition longitudinally in broiler flocks and in an experimental setting. Results from this study show that cecal droppings represented the cecal microbiota of the five individual birds across different time points quite well, even though we collected those samples in pools from the litter in the poultry house. Early in the production cycle, the composition measured with the boot socks was close to that of cecal content samples. Cecal and ileal content samples are, however, distinct from cloacal swabs. Therefore, cecal droppings and boot socks may be useful alternatives for cecal samples to determine intestinal microbiota composition longitudinally. This study contributes to expanding the toolbox for the collection of valuable data under field and experimental circumstances. With these tools, novel strategies can be developed to steer towards more resilient gut microbiota.

## 2. Materials and Methods

### 2.1. Ethical Statement

The Dutch Central Authority approved the animal experiment for Scientific Procedures on Animals and the Animal Experiments Committee (registration number AVD108002016442, 7 April 2016) of Utrecht University (Utrecht, The Netherlands), and all procedures were done in full compliance with all relevant legislation.

### 2.2. Data Collection

Data for this study were obtained from two broiler farms in The Netherlands, with Ross 308 broilers, low historic use of antibiotics, and good production performance. No antibiotics were used. Coccidiostatic drugs were applied as standard in the feed of both flocks. Both farms used a combination of nicarbazin and narasin (Maxiban^®^ G160, Elanco, Houten, The Netherlands) from day 0 onwards until day 28 followed by narasin (Monteban^®^ G100, Elanco, Houten, The Netherlands) from day 28 to day of slaughter age. At both farms water and feed were supplied ad libitum but the farms had a different feed supplier and received broilers from different commercial hatcheries. Diets on both farms were wheat based, combined with feeding of whole wheat at later ages. In addition to soybean meal, sunflower seed meal and rapeseed meal were added at maximum 6.5% inclusion.

The age of the parent stock was 55 weeks for Farm 1, and 35 weeks for Farm 2. The artificial lighting was set for 23 h/day (h/d) from days 0–3, 20 h/d for days 4–6 and 18 h/d for days 7–35. Temperature was set to gradually decrease from 34 °C at day 0 to 20 °C at day 35. Farm 1 used wood shavings and Farm 2 peat as litter material. 

During one production cycle both farms were visited on day 0 (day of arrival of the day-old chicks on the farm), 2, 7, 14 and 35. One farm was visited in August 2016, and the second farm was visited in June 2017. From each poultry house, five broilers were randomly selected, one from each corner and in the middle of the poultry house. The start of the sampling of broilers took place at least 30 min after the end of a dark period, to reduce potential effects of fasting on the amount of content in the intestinal tract and microbiota composition.

### 2.3. Sampling Methods

During each visit samples were taken from the five broilers per house before, i.e., cloacal swabs, and after euthanization, i.e., cecal content and, ileal content, and from the poultry house, i.e., cecal droppings from special feces collection boxes, and boot socks. The broilers were collected from the broiler house and immediately a cloacal swab (Rayon swab tip, Copan 155C, Copan ltalia, Brescia, Italy) was taken. The swab was placed in a 2 mL sterile cryotube (Brand™ 114832, Fisher Scientific, Wertheim, Germany) and cut with a sterile scissor to fit in the tube. Each tube with swab was placed on dry ice before being stored in a freezer (−80 °C) for future processing. Subsequently, the broilers were euthanized by cervical dislocation, immediately followed by removal of the gastrointestinal tract. This procedure was carried out in as sterile a way as possible, and between broilers sterile gloves were changed and the table, scissors, and tweezers were cleaned with 70% ethanol to prevent cross contamination between broilers. The distal end of the cecum was cut to collect cecal content. Ileal content was collected distal and close to the Meckel’s diverticulum. The intestinal content was gently squeezed into a 2 mL sterile cryotube and snap frozen on dry ice and stored at −80 °C for microbial genomic DNA extraction. The time between euthanization and placing the samples on dry ice was between 3–5 min.

In each poultry house five fecal collection boxes of wood with iron netting (70 cm × 50 cm × 5 cm) were placed on the litter, with one in each corner and one in the middle of the poultry house (Appendix A). At the start of each sample collection day, these fecal collection boxes were lined with clean oilpaper (soda paper treated with paraffin, Vendrig packaging, Woerden, The Netherlands) to collect cecal droppings. The cecal droppings were collected in the poultry house 4–6 h after placement of the boxes in the poultry house, using a sterile spoon (SteriPlast Bio, 2.5 mL, Bad Belling, Germany) or a sterile surgical disposable scalpel (Carbon Steel scalpel, #10, Braun, Tuttlingen, Germany). At least three different cecal droppings per box were put in one 2 mL cryotube and put on dry ice and stored at −80 °C until DNA extraction.

Per broiler house, at each sampling day, five boot socks (bioTRADING Benelux B.V., Mijdrecht, The Netherlands) were used to collect litter samples at the same locations where the fecal collection boxes were placed (Appendix A). Boot socks were worn over the foot while walking through a poultry house to collect adhering fecal and litter material. The poultry house boots were covered with clean plastic boot socks to avoid contamination with feces already present on the boots. Sterile gloves were used to apply the boot socks for sampling and after walking the surface of the designated location, the boot socks were placed in a plastic bag and put on dry ice and stored at −80 °C until DNA extraction.

### 2.4. DNA Extraction

In total, 50 cecal and 50 ileal content samples, 47 cecal droppings, 50 cloacal swabs and 50 boot socks were analyzed. DNA was extracted from 0.25 g cecal and ileal content and cecal droppings using 700 μL of Stool Transport and Recovery (STAR) buffer (Roche Diagnostics Nederland BV, Almere, The Netherlands). All samples were transferred to a sterile screw-capped 2 mL tube (BIOplastics BV, Landgraaf, The Netherlands) containing 0.5 g of zirconium beads (0.1 mm; BioSpec Products, Inc., Bartlesville, OK, USA) and 5 glass beads (2.5 mm; BioSpec Products) was used.

The cloacal swab samples were directly transferred to a sterile screw-capped 2 mL tube (BIOplastics BV) containing also 0.5 g of zirconium beads (0.1 mm; BioSpec Products, Inc) and 5 glass beads (2.5 mm; BioSpec Products) but 300 μL of Stool Transport and Recovery STAR buffer. 

Before DNA was extracted from a boot sock sample, the boot sock was cut in half with a sterile scalpel and put into two different 50 mL Falcon tubes. Then 35 mL phosphate-buffered saline (PBS) was added to both 50 mL tubes and the tubes were vortexed for 20 s (s) at maximum speed. Thereafter the tubes were placed for 30 min in a carousel (Shake ‘n’ Stack Thermo Hybaid, Thermo Fisher Scientific, Landsmeer, The Netherlands) to soak the feces from the boot sock (room temperature). From both tubes, 5 mL was pipetted in a 10 mL sterile tube. Next the tubes were vortexed for 5 s at maximum speed to mix and 1.5 mL of boot sock content was transferred to a sterile 2 mL microcentrifuge tube and centrifuged (13,000× *g* for 5 min at 4 °C). The supernatant was removed and of the 10 mL tube again 1.5 mL boot sock content was added and centrifuged (13,000× *g* for 5 min at 4 °C). The pellet was resuspended in STAR buffer. All samples were treated in a bead beater (Precellys 24, Bertin technologies, Montigny-le-Bretonneux, France) at a speed of 5.5 m/s for 3 × 1 min, followed by incubation at 95 °C with agitation (15 min and 300 rpm). The lysis tube was centrifuged (13,000× *g* for 5 min at 4 °C), and the supernatant was transferred to a 2 mL microcentrifuge tube. Thereafter, the aforementioned process was repeated with 300 μL STAR buffer for the cecal, cecal droppings and ileal content samples, and with 200 μL STAR buffer for the swab and boot sock samples.

An aliquot (250 μL) of the combined supernatants from the sample lysis was then transferred into the custom Maxwell^®^ 16 Tissue LEV Total RNA Purification Kit cartridge (Promega, Leiden, The Netherlands). The remainder of the extraction protocol was then carried out in the Maxwell^®^ 16 Instrument (Promega, Leiden, The Netherlands) according to the manufacturer’s instructions. The DNA concentrations were measured with a NanoDrop ND-1000 spectrophotometer (NanoDrop^®^ Technologies, Wilmington, DE, USA), and the DNA samples were stored at −20 °C until further use.

### 2.5. 16S rRNA Gene Amplification Analysis

Extracted DNA was diluted to 20 ng μL^−1^ in nuclease free H_2_O. All polymerase chain reaction (PCR) plastics were ultraviolet (UV) radiated for 15 min before use. For 16S rRNA gene-based microbial composition profiling, barcoded amplicons covering the variable regions V5-V6 of the bacterial 16S rRNA gene were generated by PCR using the 784F and 1064R primers as described in Ramiro-Garcia et al. [42].

Each sample was amplified in duplicate using Phusion hot start II high fidelity polymerase (Finnzymes, Espoo, Finland) checked for correct size and concentration on a 1% agarose gel and subsequently combined and purified using CleanNA magnetic beads (CleanN, Waddinxveen, The Netherlands). A detailed description of the PCR reactions was described previously [43].

Positive and negative controls were added to the data set to ensure high quality sequencing data. As positive controls we used synthetic communities of known composition [42], and as negative controls we used nuclease-free water. The resulting libraries were sent to GATC Biotech (Konstanz, Germany; now part of Eurofins Genomics Germany GmbH) for sequencing on an Illumina Hiseq2500 instrument. The 16S rRNA data was analyzed using the NG-tax pipeline [42]. In short, paired-end libraries were filtered to contain only read pairs with a perfect match to the primers and perfectly matching barcodes, to demultiplex reads by sample. Amplicon sequence variants (ASV) were defined as unique sequences. The ASV picking strategy was based on an open reference approach. First, reads were sorted by abundance per sample and ASVs with an abundance of <0.1% were discarded. In a second step the remaining reads were matched to the first set of ASVs with one mismatch. Taxonomy was assigned using the SILVA 128 16S rRNA gene reference database [44]. Raw sequence data were deposited into the Sequence Read Archive (SRA) at the NCBI under accession number PRJNA574842.

### 2.6. Statistical Data Analysis

To compare sample types we applied widely used diversity measures and univariate and multivariate statistical analyses. Alpha diversity (within sample) was determined using Faiths phylogenetic diversity [45] which not only takes into account the numbers of bacteria, but also the phylogenetic relatedness of those bacteria [45]. Differences in α diversity between measured microbiota by the different sample types were tested with a Kruskal–Wallis test and pairwise comparisons were tested using a Wilcoxon rank-sum test. Beta diversity (between samples) was determined using Jaccard, Bray–Curtis, weighted and unweighted UniFrac measures [46,47,48]. Multivariate microbiota data were visualized using principal coordinates analysis (PCoA), and non-parametric permutational analysis of variance (PERMANOVA) tests were used to analyze differences within multivariate community data [49]. To test for differences in relative abundance of genera between two sample types, we used a Wilcoxon rank-sum test and corrected for multiple testing with Benjamini–Hochberg (BH). We compared the cecal and ileal content to cloacal swabs, and cecal content with cecal droppings, and cecal content with boot sock samples. The comparisons between ileal content and boot socks and ileal content and cecal droppings were excluded from the data analysis because this was considered as not biologically relevant. All statistical analyses were performed in R version 3.4.2 [50], using the packages: Phyloseq, Microbiome and Vegan [51,52,53].

## 3. Results

### 3.1. Technical Results of 16S rRNA Gene Sequencing

In total, 50 cecal content, 40 ileal content, 50 cloacal swabs, 47 cecal droppings and 40 boot socks samples passed our quality control standards. The amplicon sequence variants (ASV) associated with an unknown domain or the order Mitochondria were removed from all sequenced samples. About 90% of the families in the negative controls were associated with five families: *Bacillaceae*, *Burkholderiaceae*, *Halomonadaceae*, *Micrococcaceae*, or *Shewanellaceac*. The ileal content and boot sock samples of day 0 were excluded because these contained a large number of families associated with the negative control samples, and therefore did not pass our quality control standards (Appendix A).

### 3.2. Alpha Diversity Analysis across Different Sample Types

When the data from all ages and farms were analyzed, the phylogenetic diversity was significantly higher in cecal content compared to the cloacal swab samples and the phylogenetic diversity was lower in ileal content compared to the cloacal swab samples (Figure 1, Table 1: χ^2^ = 10.6, *p* = 0.001 and χ^2^ = 18.5, *p* < 0.001). The cecal content was not significantly different from cecal droppings and boot socks based on phylogenetic diversity (Table 1: χ^2^ = 3.6, *p* = 0.058; χ^2^ = 2.6, *p* = 0.108). Alpha diversities Chao 1 and Shannon, that do not take into account the phylogenetic relatedness, showed one different result of the comparisons. According to Chao 1 and Shannon measures, the ileal content samples were not significantly different when compared to cloacal swab samples (Table 1).

### 3.3. Beta Diversity Analysis across Different Sample Types

Sample type explained 6.6%, 4.7%, 10.3% and 14.6 of the variation between the cecal content samples and cloacal swabs for Bray–Curtis, Jaccard, unweighted and weighted UniFrac distances based analysis, respectively (Table 2). For ileal content versus cloacal swabs sample, type explained 3.2%, 2.4%, 6.5%, 10.7% of the variation depending on the distance metric (Table 2). Based on unweighted UniFrac distances, no significant difference between cecal content and cecal droppings were measured, whereas on based Bray–Curtis, Jaccard and weighted UniFrac distances analysis showed that sample type explained 3.4%, 2.6% and 2.9% of the variation between the cecal content and cecal droppings. This suggests that the composition of the cecal content and cecal droppings were phylogenetically not different, but that the abundance of genera were different. The principal coordinate analysis (PCoA), in Figure 2 shows that cecal content samples and cecal droppings samples are near each other (and Appendix A with different sample types combined in single plots). Between the cecal content and boot sock samples 8.5%, 6.2%, 10.7% and 22.7% of the variation was explained by the sample type (Table 2).

### 3.4. The Effect of Age on the Bacterial Microbiota across Different Sample Types

The phylogenetic diversity of the cecal content and cecal droppings increased with the age of the broilers (Figure 1, χ^2^ = 43.8, *p* < 0.001; χ^2^ = 38.7, *p* < 0.001, Table 1). The cloacal swabs and boot sock samples also showed a significant increase in phylogenetic diversity across time but with a smaller effect size (χ^2^ = 14.9, *p* = 0.005; χ^2^ = 17.7, *p* < 0.001). No age effect was found for ileal content samples (Table 1). For the samples of day 0, cecal content showed a lower phylogenetic diversity compared to cecal droppings and cloacal swabs (Figure 1, Appendix A). On days 2, 7 and 35 no difference in the phylogenetic diversity between cecal content and cecal droppings was observed (Figure 1, Appendix A). On day 7 no differences in phylogenetic diversity were detected between cecal content and the boot socks, and between ileal content samples and cloacal swabs (Figure 1, Appendix A). Also on day 14 cecal content and boot socks were not different, but the cecal content showed a higher phylogenetic diversity compared to the cecal droppings (Figure 1, Appendix A). All other comparisons between sample types across age were significantly different (Appendix A).

For the cecal content samples the PCoA, based on unweighted UniFrac and weighted UniFrac distances, showed that the age of the broilers explained 56% and 62% of the variation (Figure 2, Table 2). When using Bray–Curtis and Jaccard distances, that do not incorporate phylogenetic distances, age explained 37.8% and 26.9% of the variation (Table 2, Appendix A). Taken together, this suggests that the taxa found in the cecal content samples were phylogenetically distinct across ages. For the ileal content samples, weighted UniFrac distances showed that age explained just 18.3% of the variation (Figure 2, Table 2). 

In the cloacal swab samples age explained 38.7% of the variation based on weighted UniFrac distances (Figure 2, Table 2), however, also β dispersion was found to be different across different ages within cloacal swabs (Table 2). Based on weighted UniFrac distances was shown that in cecal droppings age explained 68.9% of the variation (Figure 2, Table 2). In addition, the results based on weighted UniFrac distances showed that also in boot socks broiler age explained most of the variation (55.9%, Figure 2, Table 2). 

### 3.5. The Effect of Farm on the Bacterial Microbiota across Different Sample Types

No differences in phylogenetic diversity between the two farms were observed in cecal content, cecal droppings, cloacal swabs and ileal content, but differences were observed between boot sock samples from the two farms (χ^2^ = 4.8, *p* = 0.029; Table 1 and Appendix A). When comparing the different sample types on the two farms, similar results were found; the only difference was that in Farm 1 the phylogenetic diversity in cecal content was higher compared to that of cloacal swabs, while in Farm 2 this difference was not observed (*p* = 0.012 versus *p* = 0.058; Appendix A). The larger individual variation between cloacal swabs samples in Farm 2 might be the reason why we observed this difference between farms (Appendix A). 

Weighted UniFrac distances between cecal content microbiota showed that farm did not explain any of the variation (Figure 2, Table 2). When using Bray–Curtis and Jaccard distances that do not incorporate phylogenetic distances, farm explained 3% of the variation (Table 2). Taken together, this suggests that the taxa found in the cecal content samples were phylogenetically related across farms. For the ileal content samples, based on weighted UniFrac distances, results showed that 5.1% of the variation was explained by farm (Figure 2, Table 2). The Bray–Curtis and Jaccard distances also showed that farm explained 5–6% of the variation (Table 2). 

For the cloacal swab samples the results showed that farm did not explain a significant part of the variation based on unweighted and weighted UniFrac, but it did explain part of the variation based on Bray-Curtis (4.1%) and Jaccard (3.6%) (Figure 2, Table 1). For cecal droppings based on weighted UniFrac distances, showed that the farm did not explain the variation (Figure 2, Table 1). Only the Jaccard distance, taking into account the presence or absence of an ASV, showed that farm explained 3.5% of the variation (Table 1). The results based on weighted UniFrac distances showed that for boot socks, the farm explained 11.4% of the variation (Table 1). This was mainly an effect of the large difference between farms and boot socks on day 14 (Figure 2). 

### 3.6. Microbial Taxa That Differ across the Invasive and Non-Invasive Sample Types

Figure 3 shows the relative abundance across age and different sample types at family level. On day 0 both the cecal content samples and cloacal swabs were found to be predominated by members of the family *Clostridiaceae* (Figure 3). From day 2 onwards the same microbial families were highly abundant within the cecal content and cecal dropping samples (Figure 3, Appendix A). On day 2 *Enterobacteriaceae* and *Lachnospiraceae* were predominant in the cecal content, and *Lachnospiraceae* were predominant on day 7. On day 14 and day 35 *Ruminococcaceae* and *Lachnospiraceae* were the most predominant families in the cecal content samples. 

The only difference in the abundance of genera found between cecal content and cecal droppings was the increased relative abundance of the genus *Lactobacillus* in the cecal droppings compared to the cecal content on day 2 (Appendix A). The genus *Lactobacillus* was also more abundant in the cloacal swabs compared to the cecal content but also many other genera were different (Appendix A). Only on day 0, no differences between genera were observed between the cecal content and cloacal swab samples. On day 2 only the genus *Escherichia-Shigella* was higher in relative abundance in the cloacal swab samples compared to the cecal content (Appendix A). In addition, on day 2 no differences were observed between cecal content and boot socks and between ileal content and cloacal swabs.

The differences in relative abundance of genera observed in cecal content and boot socks varied across time (Appendix A). The genera *Lactobacillus* (13–17% points), *Corynebacterium* (14–28% points) and *Staphylococcus* (11–18% points) showed a higher relative abundance in the boot socks compared to cecal content samples (Appendix A). Only on day 35 did the cloacal swabs and the ileal content show differences in five genera, with a main increase in the genus *Faecalibacterium* (11% points) in the cloacal swab samples (Appendix A).

Concerning differences between farms, within the cecal content samples the only difference observed was a higher relative abundance of the genus *Bacteroides* in Farm 2 compared to Farm 1 (Appendix A, *p* = 0.005). Only in the boot socks was it also possible to observe a higher relative abundance in *Bacteroides* in Farm 2 (Appendix A, *p* = 0.029). No difference in genera between farms could be detected in cecal droppings, cloacal droppings, or ileal content samples.

## 4. Discussion

The aim of this study was to evaluate different types of samples with regard to characterization of the intestinal microbiota of individual broilers and flocks from the first week of life onwards until the end of the production cycle. In particular, we compared invasively collected cecal and ileal content samples to non-invasive samples, such as cloacal swabs, cecal droppings and boot sock samples collected from the litter. In addition, the invasive and non-invasive sample types were evaluated on the ability to distinguish between ages and among farms. The outcomes of this study suggest that cecal droppings and boot socks are useful non-invasive alternatives for cecal content samples to determine intestinal microbiota longitudinally in broilers and broiler flocks.

In our study, the cecal microbiota had the highest diversity and showed less variation over time compared to the microbiota of the ileal content, which is consistent with other studies [15,24,25]. Age had the most profound effect on the cecal microbiota as demonstrated by other authors [54,55]. Although the sample size in this study was limited to five samples per farm and time point, clear differences in microbiota across sample types were observed. It should be noted, however, that the evaluation of different sample types was based on finding no statistically significant differences between sample types. As we used non-parametric tests corrected for multiple testing, that have a low chance of false-positive results, we may have underestimated the microbiota composition differences between sample types. Therefore, we compared different methods, different α and β diversity measures and applied univariate and multivariate statistical analyses together, to determine which differences in the measured intestinal microbiota between different sample types were considered relevant.

No differences between the farms were observed in cecal content based on α diversity and β diversity taking phylogenetic relatedness into account. This is in contrast with another study where differences in microbiota composition of cecal content were found between broiler flocks [4]. Our results indicate a comparable development of the microbiota across age for both farms, despite differences between farms, such as feed supplier, bedding materials, parent stock and management. Except for these differences, both farms were similar with respect to good production performance and health status of the flock, including not using antibiotics. A comparable regimen of in-feed coccidiostat drugs in starter and grower feed, and the geographical proximity of the two farms might in part explain the absence of detectable differences in cecal microbiota composition. However, when phylogenetic relatedness was not taken into account, the farm explained a small, but significant amount of variation between the cecal content samples, indicating that the development was comparable but there were also differences in the microbiota composition between the farms. This result ties in well with a previous study wherein the choice of litter material influenced the cecal microbiota [56], which might also be the case in this study. 

When comparing non-invasive sample types to the cecal content, the cecal droppings represented the cecal microbiota of the five individual birds across different time points quite well, even though we collected those samples in pools from the litter in the poultry house. Only on day 2 of age did cecal droppings show a higher relative abundance of the genus *Lactobacillus* compared to cecal content. These results add new information to previous results described in literature, which only showed that cecal droppings reflected cecal content microbiota of broilers at the end of the production cycle [27]. Cecal droppings might be a challenge to collect, since chickens produce just one cecal dropping for every seven to eight fecal droppings [39]. Nevertheless, this sample type may be of value to monitor the cecal microbiota composition in broilers non-invasively and longitudinally. 

Another surprising potentially useful non-invasive sample type to monitor the cecal microbiota at flock level are boot socks samples. This method is commonly used in poultry houses to monitor the presence of *Salmonella* [40,41], and has recently been described to evaluate human–pathogen interactions for *Campylobacter* in the environment [57]. To our knowledge, we are the first to describe a microbiota analysis using 16S rRNA gene amplicon sequencing with boot sock samples collected in a poultry house. The composition measured with the boot socks was close to that of cecal content samples early in the production cycle. On day 2, no difference between the boot socks and cecal content samples was observed, and on day 7 only the genera *Lactobacillus*, *Escherichia-Shigella* and *Enterococcus* were lower in relative abundance in the boot socks. However, on day 35, 20 genera were different between the boot socks and the cecal content, which indicated that boot socks were less useful to reflect the cecal content microbiota near the end of the production cycle. When comparing our results to the literature, it must be pointed out that *Corynebacterium*, *Sphingobacterium*, and *Lactobacillus* were observed to be highly abundant in litter samples on day 35 [58], which indicates that the boot socks collected at the end of the production cycle may mostly represent the litter rather than the cecal or ileal content. This may suggest that the litter has developed its own specific microbiota. Within Farm 1 the composition of the boot socks at day 14 was closely related to the cecal content composition on day 35, while in Farm 2 the composition of day 14 was closely related to the cecal composition on day 7. Based on the differences we observed at day 14 between farms, we can speculate that this process of developing a litter microbiota may have occurred faster in Farm 1 than in Farm 2, potentially influenced by the difference in bedding materials with wood shavings in Farm 1 and peat in Farm 2. This is in line with the literature that showed that litter type could affect the intestinal microbiota composition [56,59].

It was expected that cecal and ileal content samples would be distinct from cloacal swabs, because that has been described before [4,21,28]. The ileal content is described to harbor a relatively simple microbial composition, with mainly the genera *Lactobacillus*, *Streptococcus* and *Clostridium* [54,60,61], compared to the rich microbial composition in the ceca [15,24,25,62]. Only on day 2 did we not observe any significant difference in the α diversity between the ileal content and cloacal swabs. Therefore, cloacal swab samples might be a useful tool early in the production round depending on the goal of the microbiota characterization.

Consistent with our present findings, previous research also showed that age explained part of the individual variation in ileal content microbiota of broilers [60]. Nevertheless, we observed that age explained only 18% of the ileal microbiota variation, whereas age explained 39% and 62% of the intestinal microbiota variation in cloacal swabs and cecal content. In contrast to the β diversity, the α diversity was not affected by age in the ileal content samples. This difference might be because ileal and cloacal microbiota are more sensitive to fluctuations of fasting or partial emptying of parts of the intestinal tract compared to cecal microbiota. This might also be the reason why the homogeneity in phylogenetic diversity between the cloacal swab samples is lower compared to homogeneity between the cecal content samples, explaining why in farm 2 the phylogenetic diversity was not significantly higher in cecal content compared to the cloacal swab samples, while all other comparisons showed significant differences. This decreases the usability of cloacal swabs for characterization of cecal or ileal microbiota based on 16S rRNA amplicon sequencing. However, major differences in the intestinal microbiota can potentially be observed with a large sample size of cloacal swabs. 

Currently, it is unclear how the intestinal microbiota can be manipulated to optimize the robustness of broiler flocks, for example, to improve the alertness of the immune system while maintaining production efficiency [63]. Knowledge about the age-related development of microbiota may contribute to future research focusing on the timing of nutritional interventions used to steer the microbiota in broiler chickens to more resilient and healthier chickens. Therefore, non-invasive longitudinal sampling methods are required in order to collect baseline samples before and after the onset of disease or a nutritional intervention. This will improve our understanding of the dynamics of intestinal microbiota in healthy individuals and those with diseases [35], and facilitates evaluation of efficacy of interventions at the individual level. Although the current high cost and time-consuming process of sequencing hampers the use of microbiota data as an applicable diagnostic tool of interventions in the field, this may change in the future. Furthermore, in experimental settings non-invasive longitudinal collection of data can already be of great value to gain more understanding of changes in intestinal microbiota composition and its associations with pathological and physiological processes in broilers. 

## 5. Conclusions

In conclusion, this study shows that the value of non-invasive sample types varies at different ages and depends on the goal of the microbiota characterization. We have shown that cecal droppings and boot socks, collected from a broiler house, are useful alternatives for cecal samples collected during post-mortem, to determine intestinal microbiota composition longitudinally in broiler flocks and in an experimental setting. These sample types may be a useful expansion of the current toolbox for microbiota studies. Further studies should be done to validate the use of those microbiota sample types as a diagnostic tool early in the production cycle, e.g., by studying broiler flocks with differences in health and productivity status in further detail. Non-invasive longitudinal sampling to monitor the development of the intestinal microbiota will facilitate the development of new and better interventions to improve the health and performance of broiler chickens.

## Figures and Tables

**Figure 1 microorganisms-07-00431-f001:**
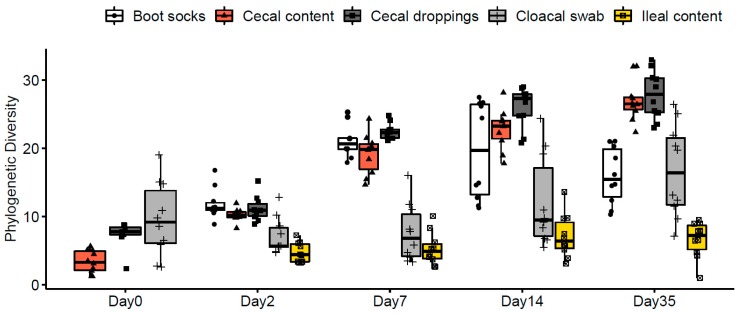
Phylogenic diversity (ASV level) of different sample types for both sampled farms across 0, 2, 7, 14 and 35 days of age. Whiskers show 95% interval, box 50% interval, *n* = 10, circles are boot socks, triangles are cecal content, squares are cecal droppings, pluses are cloacal swabs and squares with cross are ileal content samples.

**Figure 2 microorganisms-07-00431-f002:**
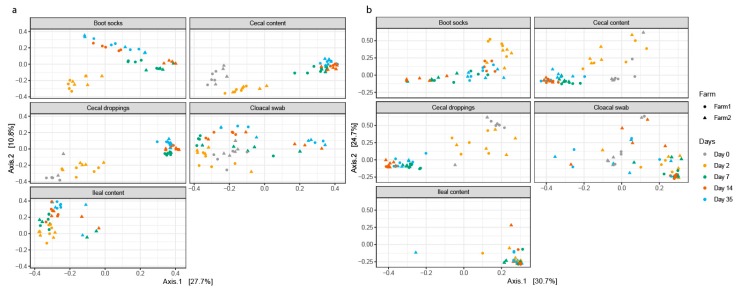
Principal coordinate plots (PCoA) based on (**a**) unweighted UniFrac and (**b**) weighted UniFrac distances between cecal and ileal content, cloacal swabs, cecal droppings and boot socks. Different sample types are shown in separate plots for clarity (Combined Appendix A). Different colors indicate different sampling days, circles are samples of Farm 1 and triangles are samples of Farm 2.

**Figure 3 microorganisms-07-00431-f003:**
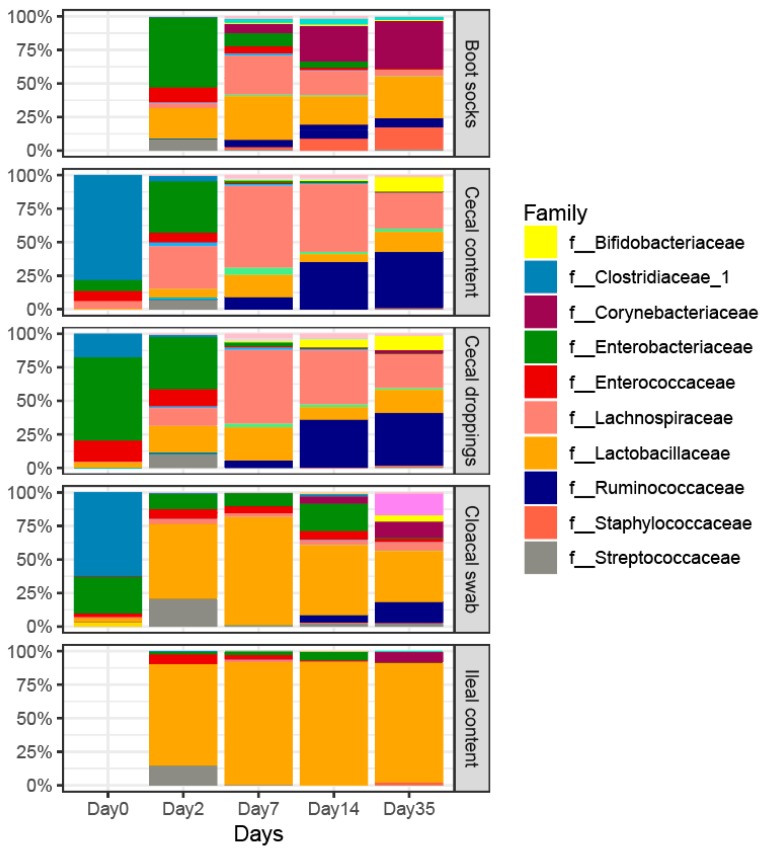
Relative microbial abundance across age and different sample types at family level. Only the labels of the 10 most abundant families are shown, the other families are not labeled for clarity. All families are presented in Appendix A.

**Table 1 microorganisms-07-00431-t001:** Different α diversity measures (ASV level) across different samples types. Number of broilers tested is specified in the table row (Kruskal–Wallis).

Sample Type		Phylogenetic Diversity		Chao 1		Shannon	
		χ^2^	*p*-Value	χ^2^	*p*-Value	χ^2^	*p*-Value
Cecal content versus cloacal swab (*n* = 100)		10.633	0.001	15.076	<0.0001	26.947	2.1 × 10^−7^
Ileal content versus cloacal swab (*n* = 80)		18.501	1.7 × 10^−5^	0.245	0.620	3.379	0.066
Cecal content versus cecal droppings (*n* = 97)		3.604	0.058	6.623	0.010	6.116	0.013
Cecal content versus boot socks (*n* = 80)		2.582	0.108	0.241	0.624	1.447	0.229
Cecal content *n* = 50	Age	43.848	6.9 × 10^−9^	38.147	1.7 × 10^−7^	37.252	1.6 × 10^−7^
	Farm	0.350	0.554	0.001	0.938	0.115	0.734
Ileal content *n* = 40	Age	3.902	0.272	3.547	0.315	11.545	0.009
	Farm	0.026	0.871	0.751	0.386	1.230	0.267
Cloacal swab n = 50	Age	14.888	0.005	13.255	0.010	13.917	0.007
	Farm	2.688	0.101	2.765	0.096	0.886	0.347
Cecal dropping *n* = 47	Age	38.679	8.1 × 10^−8^	33.46	9.4 × 10^−7^	32.865	1.3 × 10^−6^
	Farm	0.007	0.932	0.009	0.924	0.016	0.898
Boot socks *n* = 40	Age	17.729	5.7 × 10^−4^	24.527	1.9 × 10^−5^	22.945	4.1 × 10^−5^
	Farm	4.801	0.029	4.120	0.042	3.484	0.062

**Table 2 microorganisms-07-00431-t002:** Beta diversity analysis across different samples types. Overview of different distance measures that determine the microbiota interindividual diversity, R^2^ = Percentage of the variation between broilers explained, *p* = *p*-value permutational analysis of variance (PERMANOVA) test, FDis = *p*-value multivariate dispersions test.

Samples	n	Bray–Curtis			Jaccard			Unweighted UniFrac			Weighted UniFrac		
		R^2^	*p*	FDis	R^2^	*p*	FDis	R^2^	*p*	FDis	R^2^	*p*	FDis
Cecal content vs. cloacal swab	100	6.6	1 × 10^−4^	0.042	4.7	1 × 10^−4^	0.034	10.3	1 × 10^−4^	0.318	14.6	1 × 10^−4^	0.627
Ileal content vs. cloacal swab	80	3.2	0.003	0.685	2.4	0.014	0.672	6.5	2 × 10^−4^	0.569	10.7	1 × 10^−4^	0.002
Cecal content vs. cecal droppings	97	3.4	3 × 10^−4^	0.001	2.6	2 × 10^−4^	0.001	1.6	0.155	0.169	2.9	0.028	0.370
Cecal content vs. boot socks	80	8.5	1 × 10^−4^	0.001	6.2	1 × 10^−4^	0.001	10.7	1 × 10^−4^	0.626	22.7	1 × 10^−4^	0.682
Cecal content Farm	50	3.5	0.039	0.135	3.1	0.029	0.188	2.9	0.165	0.825	1.6	0.519	0.990
Cecal content Age	50	37.8	1 × 10^−4^	0.041	26.9	1 × 10^−4^	0.058	55.7	1 × 10^−4^	0.001	62.0	1 × 10^−4^	0.564
Ileal content Farm	40	6.5	0.006	0.884	5.5	0.005	0.887	3.4	0.174	0.012	5.1	0.019	0.970
Ileal content Age	40	23.0	1 × 10^−4^	0.005	18.8	1 × 10^−4^	0.007	29.6	1 × 10^−4^	0.959	18.3	1 × 10^−4^	0.245
Cloacal swab Farm	50	4.1	0.019	0.866	3.6	0.014	0.871	3.4	0.065	0.678	3.4	0.131	0.497
Cloacal swab Age	50	30.3	1 × 10^−4^	0.009	23.4	1 × 10^−4^	0.010	34.4	1 × 10^−4^	0.755	38.7	1 × 10^−4^	0.027
Cecal dropping Farm	47	3.8	0.061	0.827	3.5	0.033	0.841	3.9	0.110	0.782	2.0	0.364	0.674
Cecal dropping Age	47	42.7	1 × 10^−4^	0.937	30.5	1 × 10^−4^	0.882	64.5	1 × 10^−4^	0.006	68.9	1 × 10^−4^	0.206
Boot sock Farm	40	10.0	0.002	0.932	8.9	6 × 10^−4^	0.939	9.2	0.004	0.966	11.4	0.004	0.850
Boot sock Age	40	50.6	1 × 10^−4^	0.005	38.6	0.387	0.041	51.0	1 × 10^−4^	0.034	55.9	1 × 10^−4^	0.001

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
