# Peer review of "Comparison of Different Invasive and Non-Invasive Methods to Characterize Intestinal Microbiota throughout a Production Cycle of Broiler Chickens"

_microorganisms, 2019, doi:10.3390/microorganisms7100431_

Round 1

Reviewer 1 Report

Overall, a very enjoyable read and relevant work.

Major comments:

Missing/misplaced representations of some of the results Lines 232 to 240 only make reference to Table 1. Throughout this whole paragraph i was in want of a figure representing the alpha diversities across all samples. Then, two pages later comes Figure 2, containing exactly what i needed here. Section 3.5 (lines 294 to 317) should be accompanied by a representation of all effect sizes of the different factors (i.e. age of host, sampling location, etc.)

Minor comments:

Line 57: Should read "composition between broilers" not "composition between broilers across".

Line 230: Care to speculate as to why these samples look like negative controls? Could it be that they have very low microbial loads?

Line 245: Presenting the variation explained as an interval, enough though it's across three different methods, is confusing. Suggest you report the percentage for each, separately. 

Line 264: This statement disagrees with Figure 1.A, the ileal content panel, in which one can clearly see an age effect. 

Line 283: Directly contradicts line 264

Line 297: Is that p-value corrected for multiple testing? Looks like it wouldn't survive it.

Line 300: I know those two values look different, but i think you can confidently argue there's no effect in either.

Author Response

Reviewer report 1:

Comments and Suggestions for Authors

Overall, a very enjoyable read and relevant work.

Reply: Thank you for your positive evaluation of our manuscript and for giving us the possibility to submit a revised version. The changes have been marked in the manuscript and we provide point-by-point responses to the reviewers’ comments with reference to the line numbers in the revised version.

Major comments:

Missing/misplaced representations of some of the results Lines 232 to 240 only make reference to Table 1. Throughout this whole paragraph i was in want of a figure representing the alpha diversities across all samples. Then, two pages later comes Figure 2, containing exactly what i needed here.

Reply: We thank the reviewer for this suggestion. To improve the representation of our results we changed the order of Figures 1 and 2 accordingly (line: 236)

Section 3.5 (lines 294 to 317) should be accompanied by a representation of all effect sizes of the different factors (i.e. age of host, sampling location, etc.)

Reply: The effect sizes for all the different factors are presented in Table 1 and Table 2 (X2 and R2), but this was indeed not described in the text in the previous version of the manuscript. Therefore we added this: line 303: .. samples from the two farms (χ2 = 4.8, p = 0.029; Table 1, and Figure S4).

Minor comments:

Line 57: Should read "composition between broilers" not "composition between broilers across".

Reply: we have rephrased this sentence to, lines 56-57: In addition, there is individual variation in the broilers’ intestinal microbiota composition, both between as well as within studies and flocks.’

Line 230: Care to speculate as to why these samples look like negative controls? Could it be that they have very low microbial loads?

Independent of the sample type, all day 0 samples yielded a low amount of DNA, however, it was enough DNA to perform PCR and to sequence the samples, and this was also the case with our negative control samples. To improve the transparency of the results, the negative controls (nuclease-free water as no-template control) and all day 0 samples are presented in the supplementary file, (line 232) in Figure S2: Results of all samples collected on day 0 and the negative control samples.  a. The barplot shows the relative abundance of Bacillaceae, Burkholderiaceae, Halomonadaceae, Micrococcaceae, and Shewanellaceac in the nine negative control samples. Although overall compositional profiles were largely different from those of negative control samples (Figure S2), those families were also abundantly present (>10%) in the boot sock and ileal samples, and therefore all samples of this day were excluded for further analysis. In the cloacal swabs, the families Bacillaceae and Micrococcaceae were present, but only at relatively low abundances (3%, Table S2), and these were therefore not excluded from the analysed dataset. After day 0, those five families were still detected in some samples, but at low (>1%) abundance only (Table S2). b. This plot shows that phylogenetic diversity in samples from day 0 was higher in boot socks and ileal content than in the negative controls. This indicates that those samples are not suitable for analyses and were therefore excluded from the dataset.

Line 245: Presenting the variation explained as an interval, enough though it's across three different methods, is confusing. Suggest you report the percentage for each, separately.

Reply: We agree that presenting the data in this way may be confusing. Therefore, we agreed with your suggestion to report the percentages separately.

Lines: 247-250: Sample type explained 6.6%, 4.7%, 10.3% and 14.6 of the variation between the cecal content samples and cloacal swabs for Bray-Curtis, Jaccard, unweighted and weighted UniFrac distances based analysis, respectively (Table 2). For ileal content versus cloacal swabs, sample type explained 3.2%, 2.4%, 6.5%, 10.7% of the variation depending on the distance matrix (Table 2).

Line 253: .. showed that sample type explained 3.4%, 2.6% and 2.9% of the variation between the cecal content.

Line 258: Between the cecal content and boot sock samples 8.5%, 6.2%, 10.7% and 22.7% of the variation was explained by the sample type (Table 2).

Line 264: This statement disagrees with Figure 1.A, the ileal content panel, in which one can clearly see an age effect.

Line 283: Directly contradicts line 264

Reply: Based on the alpha diversity, there is no age effect in the ileal content samples (Line 264), while based on the beta diversity, there is an age effect (Line 283). To clarify this we added in the discussion: (Line: 435-436) In contrast to the beta diversity, the alpha diversity was not affected by age in the ileal content samples.

Line 297: Is that p-value corrected for multiple testing? Looks like it wouldn't survive it.

Reply: The p-values in Table 1 are not corrected for multiple testing. These p-values are based on the non-parametric Kruskal-Wallis test, as in Table 1 we only show comparisons between groups (e.g. farm). In Table S1, we show pairwise comparisons across sample types, age and farms. Here we have used pairwise Wilcoxon rank sum test, and these p-values were corrected for multiple testing with Benjamini-Hochberg.

To improve the manuscript we changed the description of Table S1: Alpha diversity across sample types, age and farms based on pairwise comparisons using Wilcoxon rank sum test corrected for multiple testing with Benjamini-Hochberg

Line 300: I know those two values look different, but i think you can confidently argue there's no effect in either.

Reply: In lines 234-237 “When the data from all ages and farms were analyzed, the phylogenetic diversity was significantly higher in cecal content compared to that of the cloacal swab samples and the phylogenetic diversity was lower in ileal content compared to the cloacal swab samples (Figure 1, Table 1: χ2 = 10.6, p = 0.001 and χ2 = 18.5, p < 0.001).” When we stratify per age, phylogenetic diversity on day 2, 7, 14 and 35 for the cecal content is significantly higher compared to the cloacal swab samples (Table S1, Figure S4). If we stratify the data per farm (lines 304-307, Table S1, Figure S4), this is also observed in farm 1 (p = 0.012). For farm 2 a similar trend is seen, but here with a p-value of0.058, it is slightly above the threshold to describe it as statistically significant. This is most likely mostly a result of  the especially large variation between cloacal samples (as shown in Figure S4). This large variation can be explained by the fact that cloacal swabs are more sensitive to fluctuations of fasting or partial emptying of parts of the intestinal tract compared to cecal microbiota (lines 436-347). To clarify this, we added in the discussion these lines (437-441): This might also be the reason why the homogeneity in phylogenetic diversity between the cloacal swab samples is lower compared to homogeneity between the cecal content samples, explaining why in farm 2 the phylogenetic diversity was not significantly higher in cecal content compared to the cloacal swab samples, while all other comparisons showed significant differences.

Reviewer 2 Report

Jannigje et al present a comparison of various invasive and non-invasive methods to characterize by 16S RNA sequencing the microbiota of broiler chickens. The introduction clearly presents the domain and the rationale for such study. In particular, the investigation of the usefulness of non-invasive sampling methods is key for longitudinal studies or investigations, as they do not require the post-mortem sampling. The material and method sections presents clearly the protocols followed, in particular for the new sampling methods tested, which should allow other investigator to deploy the methodologies described. The results presented suggest the potential usefulness of boot socks and cecal dropping as non-invasive methods.

The following points should however be addressed before publication.

The analysis of the data is strangely performed and justified, with a set of various methods tested and compared, until the one providing the “best” results stands up. Although such approach is fully valid and recommended to test an approach and exploit it best, the selected method should next be validated with a new set of data. The very small differences between the two farms are difficult to interpret and understand. The geographical closeness is a possibility, but the authors should present additional information (such as food composition (similarities or differences) between the two farms) to better explain such similarities. Line 226: the detection of families in the negative controls is an issue and is not considered with appropriate rigor, raising some concerns on the results presented for the samples. This information must be further detailed and the authors must explain how they used (if at all) this information in the analysis. Why are there differences in the days selected to present the various genus? This makes for example the conclusion on bacteriodes between farm 1 and farm2 (line 343) difficult to understand.

Author Response

Reviewer report 2:

Reply: Thank you for giving us the possibility to submit a revised version. The changes have been marked in the manuscript and we provide point-by-point responses to the reviewers’ comments with reference to the line numbers in the revised version.

Comments and Suggestions for Authors

Jannigje et al present a comparison of various invasive and non-invasive methods to characterize by 16S RNA sequencing the microbiota of broiler chickens. The introduction clearly presents the domain and the rationale for such study. In particular, the investigation of the usefulness of non-invasive sampling methods is key for longitudinal studies or investigations, as they do not require the post-mortem sampling. The material and method sections presents clearly the protocols followed, in particular for the new sampling methods tested, which should allow other investigator to deploy the methodologies described. The results presented suggest the potential usefulness of boot socks and cecal dropping as non-invasive methods.

The following points should however be addressed before publication.

The analysis of the data is strangely performed and justified, with a set of various methods tested and compared, until the one providing the “best” results stands up. Although such approach is fully valid and recommended to test an approach and exploit it best, the selected method should next be validated with a new set of data.

Reply: Indeed, as explained in lines 366-377 we stress that we used several approaches in the analyses to evaluate outcomes of the different sample types from different angles. We show the outcomes for a variety of sample types and were surprised to find that boot socks and cecal droppings can be added to the current toolbox for certain types of studies. It should be noted, that we do not attempt to provide a ‘best choice’ to fit all purposes. In the abstract we therefore conclude ‘This study shows that the value of non-invasive sample types varies at different ages and depends on the goal of the microbiota characterization’, suggesting that the best choice may not be the same for different types of studies.

To make this more clear, we rephrased our conclusion in lines 460-469: In conclusion, this study shows that the value of non-invasive sample types varies at different ages and depends on the goal of the microbiota characterization. We have shown that cecal droppings and boot socks, collected from a broiler house, are useful alternatives for cecal samples collected during post-mortem, to determine intestinal microbiota composition longitudinally in broiler flocks and in an experimental setting. These sample types may be a useful expansion of the current toolbox for microbiota studies. Further studies should be done to validate the use of those microbiota sample types as a diagnostic tool early in the production cycle, e.g. by studying broiler flocks with differences in health and productivity status in further detail. Non-invasive longitudinal sampling to monitor the development of the intestinal microbiota will facilitate the development of new and better interventions to improve the health and performance of broiler chickens.

We agree that our results could be validated in further research. Our goal with this paper is, however, to show the value of non-invasive sample types to microbiota characterization in broilers. We hope that other researchers will also start exploring those non-invasive sample types and can validate the usability of cecal dropping and boot sock samples, as this can help to further reduce the number of animals that need to be included in longitudinal studies.

The very small differences between the two farms are difficult to interpret and understand. The geographical closeness is a possibility, but the authors should present additional information (such as food composition (similarities or differences) between the two farms) to better explain such similarities.

Reply: We agree that we should give the readers the opportunity to have a further look into the similarities and differences by providing more information about the farms, especially about the feed. To improve the interpretation of potential difference between the farms we included additional information about the feed compositions. In line 107-109: “Diets on both farms were wheat based, combined with feeding of whole wheat at later ages. In addition to soybean meal, sunflower seed meal and rapeseed meal were added at max. 6.5% inclusion.” This indicates that the raw feed stuffs used in this study and the composition of the diets was very similar for both farms.

Line 226: the detection of families in the negative controls is an issue and is not considered with appropriate rigor, raising some concerns on the results presented for the samples. This information must be further detailed and the authors must explain how they used (if at all) this information in the analysis.

Reply: In response to your question, and to the question of Reviewer 1 about the negative controls, we have added a supplementary figure with in the caption of this figure detailed explanation on how we dealt with this in the analyses. We refer to this figure in (line 232):  supplementary Figure S2: Results of all samples collected on day 0 and the negative control samples.  a. The barplot shows the relative abundance of Bacillaceae, Burkholderiaceae, Halomonadaceae, Micrococcaceae, and Shewanellaceac in the nine negative control samples. Although overall compositional profiles were largely different from those of negative control samples (Figure S2), those families were also abundantly present (>10%) in the boot sock and ileal samples, and therefore all samples of this day were excluded for further analysis. In the cloacal swabs, the families Bacillaceae and Micrococcaceae were present, but only at relatively low abundances (3%, Table S2), and these were therefore not excluded from the analysed dataset. After day 0, those five families were still detected in some samples, but at low (>1%) abundance only (Table S2). b. This plot shows that phylogenetic diversity in samples from day 0 was higher in boot socks and ileal content than in the negative controls. This indicates that those samples are not suitable for analyses and were therefore excluded from the dataset.

Independent of the sample type, all day 0 samples yielded a low amount of DNA, however, it was enough DNA to perform PCR and to sequence the samples, and this was also the case with our negative control samples.

Why are there differences in the days selected to present the various genus? This makes for example the conclusion on bacteriodes between farm 1 and farm2 (line 343) difficult to understand.

Reply: We assume you refer to figure S5. In this figure we did not select days, but those genera were not observed in the other days. For example, for the genus Bacteriodes, you can see that it was detected on day 7, 14, and 35 for farm 1 and only detected in one sample of farm 2 at day 2.

Round 2

Reviewer 2 Report

Jannigje et al have provided rapidly constructive and clear explanations regarding the points raised for the initial manuscript. The modified version includes appropriate wordings and clarifications that justify now the manuscript to be accepted for publication.